# AnomalyHybrid: A Domain-agnostic Generative Framework for General Anomaly Detection

Ying Zhao

Ricoh Software Research Center (Beijing) Co., Ltd., China

zy_deepwhite_zy@hotmail.com

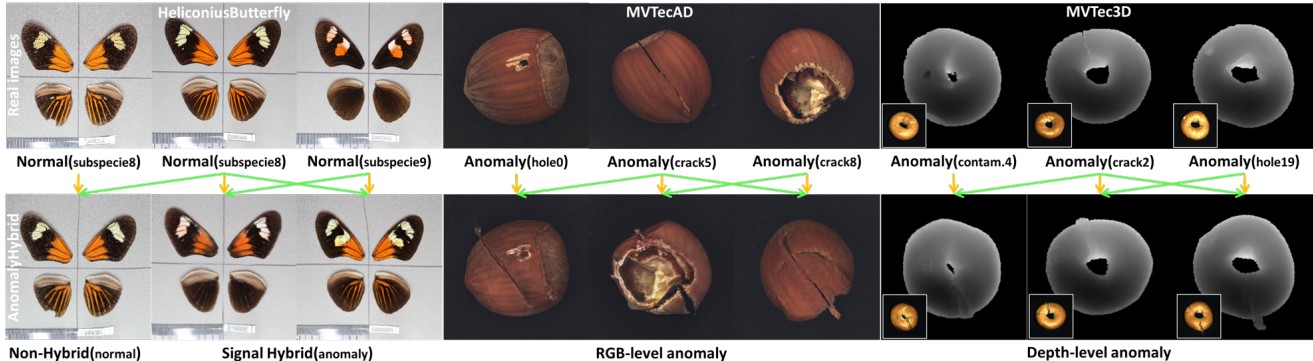

Figure 1. **AnomalyHybrid is a domain-agnostic generative framework.** Unlike prior industrial anomaly specialists, it generates general anomalies simply by combining the reference(green arrows) and target(yellow arrows) images.

## Abstract

*Anomaly generation is an effective way to mitigate data scarcity for anomaly detection task. Most existing works shine at industrial anomaly generation with multiple specialists or large generative models, rarely generalizing to anomalies in other applications. In this paper, we present AnomalyHybrid, a domain-agnostic framework designed to generate authentic and diverse anomalies simply by combining the reference and target images. AnomalyHybrid is a Generative Adversarial Network(GAN)-based framework having two decoders that integrate the appearance of reference image into the depth and edge structures of target image respectively. With the help of depth decoders, AnomalyHybrid achieves authentic generation especially for the anomalies with depth values changing, such a s protrusion and dent. More, it relaxes the fine granularity structural control of the edge decoder and brings more diversity. Without using annotations, AnomalyHybrid is easily trained with sets of color, depth and edge of same images having different augmentations. Extensive experiments carried on HeliconiusButterfly, MVTecAD and MVTec3D datasets demonstrate that AnomalyHybrid surpasses the GAN-based state-of-the-art on anomaly generation and its downstream anomaly classification, detection and segmentation tasks. On MVTecAD dataset, AnomalyHybrid achieves 2.06/0.32 IS/LPIPS for anomaly generation, 52.6 Acc for anomaly*

*classification with ResNet34, 97.3/72.9 AP for image/pixel-level anomaly detection with a simple UNet.*

## 1. Introduction

Visual anomaly detection benefits the work and economic efficiency for manufacturing industries. As anomaly is infrequent, it is barely possible to gather all kinds of real anomaly samples for training anomaly detectors. The performance of anomaly detectors are greatly constrained by the scarcity of real anomalies. Besides that, the normal appearance of a same product can also varies from sample to sample. With limited training data, it is challenging to construct a robust anomaly detector to handle unseen cases. The emergence of model-free anomaly synthesis [5, 14, 21, 28, 30, 40] and model-based anomaly generation methods [9, 11, 33, 38, 39] has catalyzed significant strides in anomaly detection.

The model-free anomaly synthesis methods [5, 14, 28, 30] are basically based on image processing paradigm of fusing selected anomaly regions to the normal image. They mainly differ in strategies of region selection, anomaly sourcing and image fusion. While evolving with different fusion strategies [36, 37], the synthetic anomalies produced by model-free methods are far from realistic. The model-based anomaly generation methods output more re-

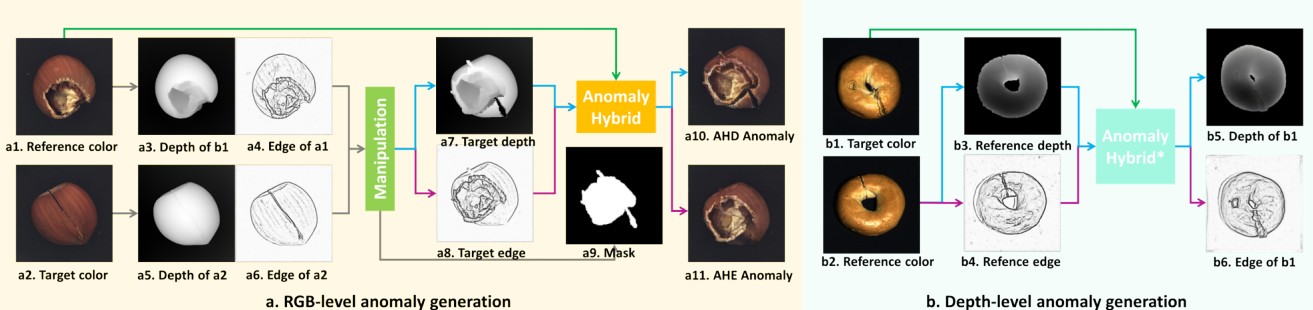

Figure 2. **Inference workflow of AnomalyHybrid.** AnomalyHybrid combines the appearance of reference image with the depth and edge structural target image. It generates global and local anomalies without and with applying manipulations on target depth and edge maps.

alistic samples by employing generative models, like Generative Adversarial Networks (GANs) and Diffusion models, in supervised [9, 11] and unsupervised [33, 38, 39] ways. Though effective, the supervised methods barely can generate unseen anomaly types. On the other hand, the unsupervised generative methods focus only on image-level anomaly generation but ignore the depth-level. They are still struggling to generate realistic anomalies along with depth values changing, such as protrusion and dent. Besides generation quality, efficiency and generalization capability of anomaly generation methods are also worthy of attention. With increasing amounts of objects from versatile datasets, it is infeasible to learn multiple large and dedicated generative models per category or dataset.

To solve aforementioned problems, we propose AnomalyHybrid that is a simple framework designed to generate diversity and authentic anomalies across application domains with color, depth and edge conditional controls. It is a GAN-based framework having two decoders that integrate the appearance of reference image into the depth and edge structures of target image respectively. To demonstrate the generation quality and generalization ability of AnomalyHybrid, Fig.1 visualizes the results produced by models trained on 3 datasets across application domains. For HeliconiusButterfly dataset, the anomaly is the hybrid butterfly of two non-hybrid subspecies. The hybrid butterfly contains appearance information of its parents. On the contrary, for the industrial anomaly datasets, such as MVtecAD and MVtec3D, the anomaly is local regions having different in depth or color values, such as crack and hole anomalies. Without network structure modification, the proposed AnomalyHybrid can easily transfer to generate anomalies for these different application domains.

In summary, we make following main contributions:

- We propose a domain-agnostic framework, AnomalyHybrid, that not only works for industrial anomaly scenarios but also can be easily transfer to generate anomalies for broad applications. Experiments carried on industrial and biological datasets valid its generalization ability.
- AnomalyHybrid consists of depth and edge decoders that substantiate each other to generates diverse and au-

thentic both normal and anomaly images. We achieve 2.06/0.32 and 1.85/0.24 IS/LPIPS for anomaly generation on MVTecAD and MVTec3D, that is better than recent GAN-based and diffusion-based SOTA.

- We conduct extensive experiments to demonstrate that our generated images bring benefits to downstream anomaly detection tasks. On MVTecAD dataset, we achieve 52.6 Acc for anomaly classification with ResNet34, 97.3/72.9 AP for image/pixel-level anomaly detection with a simple UNet, that surpasses the GAN-based SOTA.

## 2. Related works

**Image-level Anomaly Synthesis.** With the merits of simple and efficient, anomaly synthesis is widely used in unsupervised anomaly detection methods [14, 21, 28, 36, 37, 40]. CutPaste [14] syntheses anomalies by creating local discontinuous regions with the cut-paste processing. It cuts local rectangular regions from normal images and directly paste them back at random positions. SPD [40] and NSA [21] improve it by adding different strategies to smooth the pasting boundary. To increase diversity of synthetic anomaly, Draem [28] extracts anomaly source from an extra DTD [6] dataset in irregular regions obtained by using binarized Perlin noise. To make the synthesis more naturally, JNLD [36] simulates different levels of anomalies based on the just noticeable distortion [25]. OmniAL [37] extends JNLD [36] by controlling the portion of synthetic anomalies with a panel-guided strategy.

**Depth-level Anomaly Synthesis.** Recent methods flourish the Perlin noise based anomaly synthesis paradigm of Draem [28] from various aspects, such as EasyNet [5], DBRN [3], 3DSR [30] and 3Draem [29] extend it to depth-level anomaly synthesis. EasyNet [5] takes the Perlin noise as the anomaly depth values and injects them to the selected regions of depth image to produce depth-level anomaly. Meanwhile, it regards random texture as the anomaly image values and uses the same Perlin noise to guide the image-level anomaly synthesis. Slight differently, DBRN [3] simulates the anomaly depth values by normalizing the

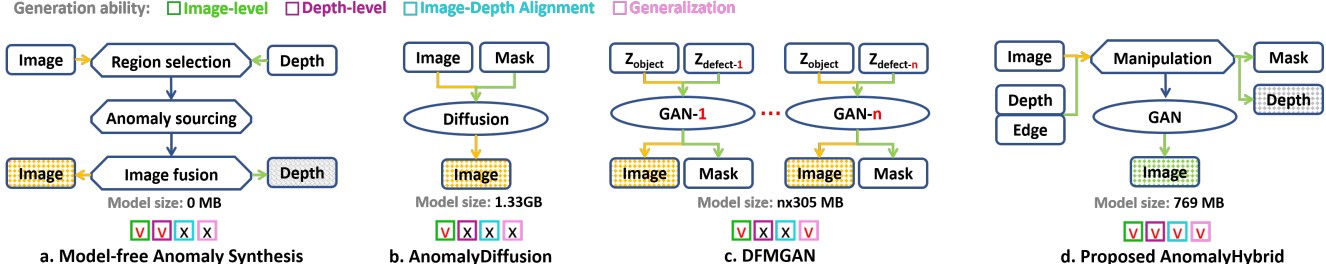

Figure 3. **Comparison of related frameworks.** (a) summarizes the three key components in model-free anomaly synthesis methods, such as Draem [28] and 3DSR [30]. (b) relies on large diffusion model. (c) achieves authentic anomaly generation by learning multiple defect-aware specialists. Comparing to previous workflows, (d) our proposed AnomalyHybrid has more comprehensive generation ability.

same texture image used for image-level anomaly synthesis. Based on the handcrafted principles of anomaly depth values, changing gradually, capturing local changes and variable average object depth, 3DSR [30] forges the depth-level anomaly by adapting the Perlin noise image with a randomized affine transform. Similarly, 3Draem [29] uses the Perlin noise generator to create anomaly regions and smooths the simulated depth values to ensure more consistent local depth changes. Though the synthetic image-level and depth-level anomalies are in same location, there is no guarantee that the texture and depth have the same changing tendency. Without considering alignment of depth-level and image-level information, these methods usually generate unrealistic anomalies.

**Image-level Unsupervised Anomaly Generation.** RealNet [33] proposes a diffusion process-based synthesis strategy that generates anomaly samples by blending the normal image with the diffusion generated anomalous texture. To mimic real anomalies distribution, it carefully selects the parameter that controls the strength of anomaly generation. GRAD [7] proposes a diffusion model to generate both structural and logical anomaly patterns. It generates contrastive patterns by preserving the local structures while disregarding the global structures present in normal images. Moreover, it uses a self-supervised re-weighting mechanism to handle the challenge of long-tailed and unlabeled synthetic contrastive patterns. LogicAL [39] proposes to generate logical and structural anomalies with a GAN-based framework by manipulating edges in semantic and arbitrary regions. AnomalyFactory [38] designs a GAN-based network architecture that combines structure of a target map and appearance of a reference color image with the guidance of a learned heatmap. It has strong scalability in generating various types of samples with anomaly heatmaps for training an unified anomaly predictor for multiple categories of different datasets. Due to lack of depth information, these methods barely generate anomalies having realistic depth changing, such as protrusion and dent.

**Image-level Supervised Anomaly Generation.** To obtain realistic anomalies, more and more methods [9, 11, 18, 31] are equipped with the powerful generative models, like GANs and Diffusion models. SDGAN [18] generates surface defects with GANs trained by cycle consistency loss on a small number of real defect images. DefectGAN [31] generates realistic defect samples with GANs by simulating the defacement and restoration processes with a layer-wise composition strategy. DFMGAN [9] attaches defect-aware residual blocks to the pre-trained StyleGAN2 [13] backbone and manipulates the features within the learnt defect masks. AnoDiffusion [11] proposes a diffusion-based few-shot anomaly generation model that separately learns the anomaly appearance and location information, then generates the anomaly on the masked normal samples. These supervised anomaly generation methods, though effective, rely on real anomalous images and cannot generate unseen anomaly types.

**Depth-to-image generation.** According to recent survey [17] on multimodal unsupervised anomaly detection, there is no method to generate depth-level anomaly with generative models. We further investigate generation methods of depth-to-image. To enhance the controllability of pre-trained text-to-image diffusion models, many efforts [15, 20, 32, 35] focus on incorporating it with image-based conditional controls, such as depth map. UniControl [20] introduces a mixture of expert (MOE)-style adapter and a task-aware HyperNet to modulate the diffusion models, enabling the adaptation to different condition-to-image tasks simultaneously. Uni-ControlNet [35] leverages two lightweight adapters to enable local and global controls over pre-trained text-to-image diffusion models. With shared local and global condition encoder adapters, it injects multiscale local condition and concatenates global visual conditional tokens with text tokens respectively. ControlNet [32] proposes a neural network architecture to add spatial conditioning controls to large, pretrained text-to-image diffusion models. The neural architecture is connected with zero-initialized convolution layers that progressively grow the parameters from zero and ensure that no harmful noise could affect the fine-tuning. ControlNet++ [15] improves controllable generation of ControlNet [32] by explicitly optimizing pixel-level cycle consistency between generated images and conditional controls.

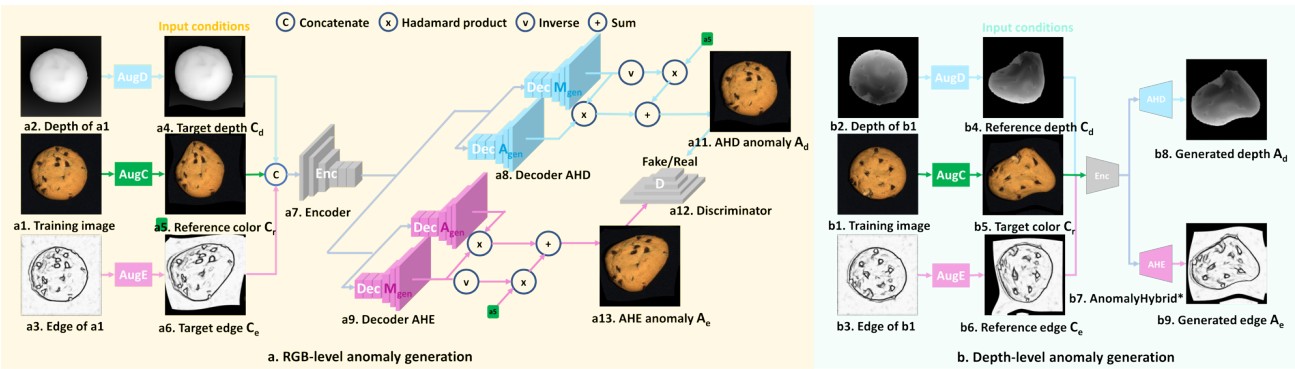

Figure 4. **Training workflow of AnomalyHybrid.** AnomalyHybrid is trained with sets of depth, color, edge of same images but having different augmentations. It consists of an encoder, two decoders and a discriminator. All decoders consist of anomaly texture and mask branches. The two-branch architecture forces the network to inject the appearance of reference to the structural of target depth and edge.

As summarized in Fig.3, different with the aforementioned methods, such as [28, 30, 33, 38], we propose a GAN-based framework that generates and aligns different levels of anomalies across versatile application domains.

## 3. Methods

### 3.1. Overview

AnomalyHybrid has a GAN-based network architecture. Fig.4 demonstrates the training workflow of AnomalyHybrid for both RGB-level and depth-level anomaly generation. Without using annotations, AnomalyHybrid is trained in an unsupervised way using sets of RGB, depth and edge of the same images with different augmentations. Since most datasets contain only RGB images, the edge and depth maps are extracted by pre-trained PiDiNet [22] and DepthAnythingV2 [26] respectively. As shown in Fig.4a, during training phase, AnomalyHybrid learns to generate RGB images for the target depth and edge maps conditioning on the reference RGB images. Thanks to the heavy augmentations, AnomalyHybrid learns to convert any depth and edge maps into RGB images that share appearance of the reference RGB images. It brings benefits that AnomalyHybrid can generate local anomalies by simply manipulating the edge and depth maps during inference phase, as shown in Fig.2a. As illustrated in Fig.4b, the depth-level anomaly generator, AnomalyHybrid*, is trained in a similar way but with different learning targets. It learns to extract depth and edge maps for the target RGB images referring to the input depth and edge maps. With AnomalyHybrid and AnomalyHybrid*, we can get aligned RGB-level and depth-level anomalies for 3D datasets, such as MVTec3D [2] demonstrated in Fig.2b.

### 3.2. Network architecture

AnomalyHybrid's network architecture is motivated by the observations of task representation and data flow. Anomaly generation $A_{out}$, in different levels, can be generally taken as a fusion of generated anomaly source $A_{gen}$ and input reference content $C_{in}$ under the guidance of a generated fusion map $M_{gen}$. It can be defined as following formulation.

$$A_{out} = C_{in} \cdot (1 - M_{gen}) + A_{gen} \cdot M_{gen} \qquad (1)$$

To generate diverse and authentic anomaly source $A_{gen}$, we consider to simultaneously use depth and edge conditions to control local structure and image condition to control global appearance. Therefore, we design a GAN-based network architecture shown in Fig.4.

Generally, AnomalyHybrid follows the encoder-decoder architecture that is broadly used in conditional generative adversarial network (cGAN) models, such as pix2pixHD [24]. It mainly evolves four scales features encoding and decoding with basic convolution blocks and ResnetBlocks. The encoder extracts multi-scale features of concatenated conditions of image, depth and edge. Two groups of dedicated decoders, AHD and AHE, target to translate the encoded features into generations that are controlled by the depth and edge conditions respectively. Each group of decoders has two branches to generate the anomaly source $A_{gen}$ and fusion map $M_{gen}$ corresponding to Eq.1. The fusion results, AHD and AHE anomalies, are fed into a discriminator to distinguish the generation quality comparing to the real inputs.

### 3.3. Training data preparation

**Edge.** We extract edge maps with pre-trained PiDiNet [22] that is one of the appealing edge detectors that achieve a better trade-off between accuracy and efficiency. It integrates the advantages of traditional edge detectors and deep CNNs by using the well-designed pixel difference convolution. By learning from different annotations, it can produce four granularities of edge maps in which the first one contains the most details. As shown in the bottom of Fig.4a3 and Fig.4b3, the first granularity edge maps of PiDiNet [22] mainly contain high-level semantic edges, such as contours of deer, riverside and forest. Therefore, we take only the first granularity edge maps as our edge condition controls.

**Depth.** We estimate depth maps also with the pre-trained model, DepthAnythingV2 [26]. It is a powerful foundation model for monocular depth estimation. It produces robust predictions for complex scenes with fine details. Comparing to previous methods, its most critical modification is replacing all labelled real images with precise synthetic images. It overcomes the drawbacks of using real labelled images that contain noise and overlooks certain details in depth maps. Following this guidance, we use pseudo depth maps extracted by DepthAnythingV2 [26] for all datasets in RGB-level anomaly generation. However, the pseudo depth maps is much less accurate than the real ones, as demonstrated in Fig.4a2 and Fig.4b2. Therefore, we use the real depth maps in depth-level anomaly generation for MVTec3D [2] dataset.

### 3.4. Unsupervised training

Without using annotations, we construct sets of image, depth and edge conditions having different augmentations for training. Generally, the input conditions are applied different augmentations and the generate contents share the same augmentations with the target conditions. The consistency of augmentations are indicated with same color shown in Fig.4. As illustrated in Fig.4a, the generated AHD and AHE anomalies are applied same augmentations with the target depth and edge conditions accordingly.

**Augmentation.** Following [23, 38, 39], our augmentations mainly consist of local thin-plate-spline (TPS) [8] warps, resize-translation-padding and top-bottom/left-right flip. The local TPS randomly shifts 3x3 control points from a local region in the horizontal and vertical directions. Compare to selecting control points globally, the local TPS brings smaller spatial range of warps. The resize-translation-padding augmentation is mainly design to counter the drastically edge manipulation on texture categories, such as editing edges of the most parts of Hazelnut [1] category. The flip augmentation brings the benefits of direction-agnostic authentic generation. By using these three augmentations, we get a generator that is robust to the drastic manipulations, as shown in Fig.5.

**Losses.** Following [23, 38, 39], we use the VGG perceptual loss [12] $L_{vgg}$ to measure the fidelity of generation $G(C_x, A_y)$ and the conditional GAN loss $L_D$ to measure the differentiate between the generated and true images. The loss of anomaly generation $L_A$ is defined as follow.

$$L_G(C_x, A_y; G) = L_{vgg}(G(C_x), A_y) \qquad (2)$$

$$L_D(C_x, A_y; D, G) = log(D(C_x, A_y))$$
$$+ log(1 - D(C_x, G(C_x))) \qquad (3)$$

$$L_A = L_G(C_x, A_y; G) + L_D(C_x, A_y; D, G) \qquad (4)$$

Where, $C_{x=\{d,r,e\}}$ indicate the input depth $C_d$, color $C_r$ and edge $C_e$ conditions, $A_{y=\{d,e\}}$ denote the target images $A_d$ and $A_e$ for AHD and AHE anomaly generations, $G$ is the generator and $D$ is the discriminator.

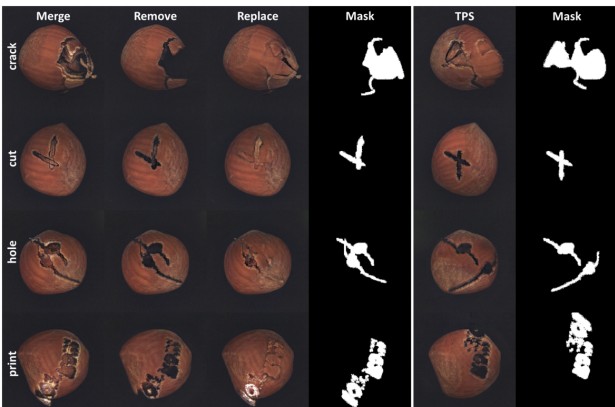

Figure 5. **Examples of anomaly generation using different manipulations on Hazelnut of MVTecAD [1].**

### 3.5. Anomaly generation

As shown in Fig.1, AnomalyHybrid can generate global and local anomalies for different applications. For global anomaly generation, it directly combines appearance and structure features of the reference and target images, such as the butterfly hybrid. As demonstrated in Fig.7, the AHD and AHE decoders bring diverse generations by focusing on depth and edge controlled butterfly hybrid(anomaly) respectively. In terms of local anomaly generation, AnomalyHybrid applies manipulations on depth and edge maps to make the anomalies more diverse and authentic. Fig.2a summarizes the workflow of generating anomalies by applying local manipulations on the depth and edge maps before feeding them to the generator.

Following [38, 39], the basic manipulation consists of simply removing, replacing, merging conditions and applying TPS on conditions in local regions.

- **Mask.** The local anomaly regions are the combination of augmented anomaly regions of reference and target images. The augmentation basically consists of resize, flip and crop.
- **Merge.** The depth and edge values in the anomaly regions of reference and target images are merged together to forge new anomaly textures.
- **Remove.** To forge the typical defects, such as crack, cut and hole, the depth and edge values in the anomaly regions are set as background values.
- **Replace.** To reduce the intensity level of manipulation, the depth and edge values in the anomaly regions of target image are replaced by the values in the reference image.
- **TPS.** To increase anomaly diversity, we apply TPS on anomaly regions and get various anomaly textures.

Fig.5 and Table 7 illustrate the anomaly generations with different manipulations for four types of Hazelnut defects.

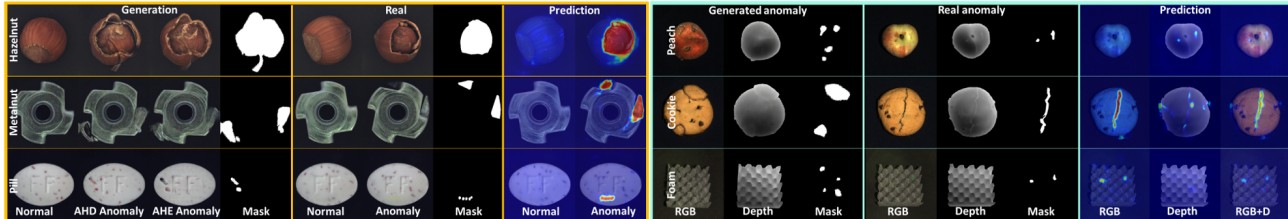

Figure 6. **Examples of anomaly generation and detection by AnomalyHybrid on (Left)MVTecAD [1] and (Right)MVTec3D [2].**

Table 1. Comparison of anomaly generation and classification performance using same ResNet34 on MVTecAD [1]. (AnoDiffusion is excluded for ranking, denoted as gray, since it trains classifiers with selected generation images.)

| Category (NO.defects) | DiffAug[34] | CropPaste[16] | SDGAN[27] | DGAN[31] | DFMGAN[9] | **AnomalyHybrid** | AnoDiffusion[11] |
|---|---|---|---|---|---|---|---|
| | GAN-based | | | | | | Diffusion-based |
| | IS↑ (Inception Score) / LPIPS↑ (Learned Perceptual Image Patch Similarity) / Classification Accuracy↑ | | | | | | |
| bottle(3) | 1.59/0.03/48.8 | 1.43/0.04/52.7 | 1.57/0.06/48.8 | 1.39/0.07/53.5 | 1.62/0.12/56.6 | **2.01/0.23/62.5** | 1.58/0.19/90.7 |
| cable(8) | 1.72/0.07/21.4 | 1.74/0.25/32.8 | 1.89/0.19/21.9 | 1.70/0.22/21.4 | 1.96/0.25/**45.3** | **2.75/0.42**/41.1 | 2.13/0.41/67.2 |
| capsule(5) | 1.34/0.03/34.7 | 1.23/0.05/32.9 | 1.49/0.03/30.2 | 1.59/0.04/32.0 | 1.59/0.11/37.2 | **2.33/0.29/40.0** | 1.59/0.21/66.7 |
| carpet(5) | 1.19/0.06/35.5 | 1.17/0.11/28.0 | 1.18/0.11/21.5 | 1.24/0.12/29.0 | 1.23/0.13/**47.3** | 1.43/**0.29**/33.3 | 1.16/0.24/58.1 |
| grid(5) | 1.96/0.06/28.3 | 2.00/0.12/28.3 | 1.95/0.10/30.8 | **2.01**/0.12/27.5 | 1.97/0.13/**40.8** | 1.92/**0.28**/40.0 | 2.04/0.44/42.5 |
| hazelnut(4) | 1.67/0.05/65.3 | 1.74/0.21/59.0 | 1.85/0.16/43.8 | 1.87/0.19/61.1 | 1.93/0.24/**81.9** | **2.16/0.33**/77.6 | 2.13/0.31/85.4 |
| leather(5) | 2.07/0.06/40.7 | 1.47/0.14/34.4 | 2.04/0.12/38.1 | 2.12/0.14/42.3 | 2.06/0.17/49.7 | **2.56/0.38/62.1** | 1.94/0.41/61.9 |
| metal nut(4) | 1.58/0.29/58.9 | 1.56/0.15/60.0 | 1.45/0.28/44.3 | 1.47/0.30/56.8 | 1.49/**0.32**/64.6 | **1.88**/0.27/**68.3** | 1.96/0.30/59.4 |
| pill(7) | 1.53/0.05/29.9 | 1.49/0.11/26.7 | 1.61/0.07/20.5 | 1.61/0.10/28.5 | 1.63/0.16/29.5 | **2.06/0.26/43.8** | 1.61/0.26/59.4 |
| screw(5) | 1.10/0.10/25.1 | 1.12/0.16/28.8 | 1.17/0.10/26.8 | **1.19**/0.12/28.8 | 1.12/0.14/**37.5** | 1.14/**0.24**/34.2 | 1.28/0.30/48.2 |
| tile(5) | 1.93/0.09/59.7 | 1.83/0.20/68.4 | 2.53/0.21/42.7 | 2.35/0.22/26.9 | 2.39/0.22/74.9 | **2.89/0.49/88.5** | 2.54/0.55/84.2 |
| toothbrush(1) | 1.33/0.06/- | 1.30/0.08/- | 1.78/0.03/- | 1.85/0.03/- | 1.82/0.18/- | **1.95/0.25**/- | 1.68/0.21/- |
| transistor(4) | 1.34/0.05/38.1 | 1.39/0.15/41.7 | 1.76/0.13/32.1 | 1.47/0.13/35.7 | 1.64/0.25/**52.4** | **2.13/0.41**/45.8 | 1.57/0.34/60.7 |
| wood(5) | 2.05/0.30/41.3 | 1.95/0.23/47.6 | 2.12/0.25/31.0 | **2.19**/0.29/24.6 | 2.12/0.35/49.2 | 2.09/**0.38/64.9** | 2.33/0.37/71.4 |
| zipper(7) | 1.30/0.05/22.8 | 1.23/0.11/26.4 | 1.25/0.10/21.5 | 1.25/0.10/18.7 | 1.29/**0.27**/27.6 | **1.65**/0.25/**34.7** | 1.39/0.25/69.5 |
| Mean | 1.58/0.09/39.3 | 1.51/0.14/40.6 | 1.71/0.13/32.4 | 1.69/0.15/34.8 | 1.72/0.20/49.6 | **2.06/0.32/52.6** | 1.80/0.32/66.1 |

## 4. Experiments

### 4.1. Datasets

We conduct extensive experiments on 2 industrial datasets, MVTecAD [1] and MVTec3D [2], and 1 biological dataset, HeliconiusButterfly [4]. **MVTecAD** [1] contains 3,629 high-resolution color images from 15 different categories of industrial objects and textures in trainset. Its testset contains 70 types of structural anomalies in different categories, including broken, crack, contamination and misplacement. **MVTec3D** [2] contains over 4,147 high-resolution scans of 10 categories acquired by an industrial 3D sensor that acquires RGB data. There are 894 anomalous containing various defects that are visible in either RGB or 3D data. **HeliconiusButterfly** [4] contains high-resolution (5184x3456) images of non-hybrid (normal) and hybrid (anomaly) subspecies of Heliconius butterfly. The trainset contains 2,084 images of 14 non-hybrid and 1 hybrid subspecies. The testset has 2,350 images of 16 non-hybrid and 7 hybrid subspecies. According to the number of hybrid subspecies, it split them into the Signal Hybrid and Non-Signal Hybrid. The unseen hybrid in testset is called as Mimic Hybrid. The visual appearances (e.g., color patterns on the wings) of these subspecies can be drastically different. More details are shown in the Supplementary.

### 4.2. Metrics

**Anomaly generation** Following [11, 38], we utilize Inception Score(**IS**) and cluster-based Learned Perceptual Image Patch Similarity(**LPIPS**) to evaluate the realistic and diversity of our generation. IS measures the realistic and diversity of generated images but is independent of the given real anomaly data. A higher IS indicates better realistic and greater diversity. LPIPS computes the similarity between the features of two image patches extracted from a pre-trained network. The higher LPIPS the greater variety generated images are.

**Anomaly detection** For butterfly hybrid detection, we use the harmonic mean of the Signal Hybrid Recall, Non-Signal Hybrid Recall, and Mimic Hybrid Recall as the final score. The true positive rate (TPR) at the true negative rate (TNR) is set as 0.95. That is recall of hybrid cases, with a score threshold set to recognizing non-hybrid cases with 0.95 accuracy. For industrial anomaly inspection, we utilize AUROC, Average Precision (AP), and the F1-max score to evaluate the accuracy of image-level anomaly detection and pixel-level anomaly localization.

### 4.3. Main results

**Anomaly generation.** Following previous works[9, 11], we randomly choose 1/3 of the dataset images from each defect

Table 2. Comparison of anomaly localization and detection performance using same UNet on MVTecAD [1]. AnoHybrid trains UNet with images generated by both depth and edge decoders. AnoHybrid+ indicates additionally using generated normal images for training.

| Category | CropPaste[16] | DFMGAN[9] | AnoHybrid | AnoHybrid+ | CropPaste[16] | DFMGAN[9] | AnoHybrid | AnoHybrid+ |
|---|---|---|---|---|---|---|---|---|
| | Pixel-level AUC/AP/$F_1$-max | | | | Image-level AUC/AP/$F_1$-max | | | |
| bottle | 94.5/67.4/63.5 | 98.9/90.2/83.9 | 98.3/77.2/72.5 | 98.5/78.7/74.6 | 85.4/95.1/90.9 | 99.3/99.8/97.7 | 99.2/99.8/98.7 | 99.6/99.9/98.7 |
| cable | 96.0/75.3/69.3 | 97.2/81.0/75.4 | 94.1/76.0/73.4 | 93.0/75.5/72.3 | 93.3/96.1/91.6 | 95.9/97.8/93.8 | 96.1/97.7/91.9 | 97.3/98.5/94.4 |
| capsule | 95.3/49.2/51.1 | 79.2/26.0/35.0 | 98.4/51.7/54.6 | 97.3/44.5/49.1 | 77.1/94.1/90.4 | 92.8/98.5/94.5 | 94.9/98.8/95.2 | 89.0/97.6/93.5 |
| carpet | 83.7/36.6/39.7 | 90.6/33.4/38.1 | 98.6/82.8/75.6 | 98.5/82.3/76.2 | 57.7/84.3/87.3 | 67.9/87.9/87.3 | 96.3/98.9/94.0 | 97.6/99.3/96.4 |
| grid | 84.7/13.1/22.4 | 75.2/14.3/20.5 | 98.8/58.6/59.2 | 98.7/59.9/59.4 | 83.0/94.1/87.6 | 73.0/90.4/85.4 | 100/100/100 | 100/100/100 |
| hazelnut | 88.5/38.0/42.8 | 99.7/95.2/89.5 | 99.6/89.4/82.6 | 99.5/84.8/79.3 | 68.8/85.0/78.0 | 99.9/100/99.0 | 96.7/98.3/92.6 | 93.8/97.1/91.8 |
| leather | 97.5/76.0/70.8 | 98.5/68.7/66.7 | 99.6/72.7/67.1 | 99.4/67.9/64.9 | 91.9/97.5/90.9 | 99.9/100/99.2 | 98.4/99.5/97.4 | 99.3/99.7/98.3 |
| metal nut | 96.3/84.2/74.0 | 99.3/98.1/94.5 | 98.8/93.5/87.0 | 98.6/93.0/87.5 | 92.2/98.1/93.3 | 99.3/99.8/99.2 | 99.8/99.9/99.2 | 99.8/99.9/99.2 |
| pill | 81.5/17.8/24.3 | 81.2/67.8/72.6 | 99.3/94.9/88.4 | 99.5/94.9/88.0 | 51.7/87.1/91.4 | 68.7/91.7/91.4 | 99.1/99.8/98.9 | 98.3/99.7/97.2 |
| screw | 93.4/31.2/36.0 | 58.8/2.2/5.3 | 77.0/7.8/6.4 | 74.3/3.9/11.1 | 59.3/81.9/86.0 | 22.3/64.7/85.3 | 44.6/72.6/84.9 | 50.4/75.6/85.2 |
| tile | 94.0/79.3/74.5 | 99.5/97.1/91.6 | 99.3/94.6/87.4 | 99.3/94.5/86.4 | 73.8/91.1/83.8 | 100/100/100 | 99.5/99.8/99.0 | 99.6/99.9/98.1 |
| toothbrush | 89.3/30.9/34.6 | 96.4/75.9/72.6 | 98.7/65.2/67.8 | 98.2/64.5/67.0 | 81.2/91.0/88.9 | 100/100/100 | 100/100/100 | 100/100/100 |
| transistor | 85.9/52.5/52.1 | 96.2/81.2/77.0 | 98.1/80.8/74.2 | 96.8/73.8/70.2 | 85.9/81.8/80.0 | 90.8/92.5/88.9 | 92.9/90.4/86.3 | 95.2/93.4/86.4 |
| wood | 84.0/45.7/48.0 | 95.3/70.7/65.8 | 95.8/70.7/64.8 | 96.4/73.1/66.6 | 49.5/81.2/86.6 | 98.4/99.4/98.8 | 96.6/98.7/98.7 | 96.6/98.8/97.3 |
| zipper | 94.8/47.6/51.4 | 92.9/65.6/64.9 | 99.1/82.3/74.9 | 98.9/81.7/73.7 | 59.4/82.8/88.9 | 99.7/99.9/99.4 | 99.9/100/99.3 | 100/100/100 |
| Mean | 90.4/48.4/49.4 | 90.0/62.7/62.1 | 96.9/72.9/69.1 | 96.5/71.5/68.4 | 74.0/89.4/87.7 | 87.2/94.8/94.7 | 94.3/96.9/95.7 | 94.4/97.3/95.8 |

Table 3. Comparison of anomaly generation on MVTec3D [2].

| Category | DFMGAN[9] | AnoDiffusion[11] | AnomalyHybrid | |
|---|---|---|---|---|
| | RGB-level | | | Depth-level |
| | IS ↑/LPIPS ↑ | | | |
| bagel | 1.07/0.26 | 1.02/0.22 | 1.05/0.23 | 1.52/0.14 |
| cableG | 1.59/0.25 | 1.79/0.19 | 2.42/0.21 | 2.63/0.21 |
| carrot | 1.94 /0.29 | 1.66/0.17 | 2.31/0.21 | 2.02/0.11 |
| cookie | 1.80/0.31 | 1.77/0.29 | 1.95/0.28 | 1.45/0.16 |
| dowel | 1.96/0.37 | 1.60/0.20 | 1.89/0.22 | 1.78/0.15 |
| foam | 1.50/0.17 | 1.77/0.30 | 1.73/0.28 | 1.36/0.19 |
| peach | 2.11/0.34 | 1.91/0.23 | 1.97/0.25 | 1.71/0.13 |
| potato | 3.05/0.35 | 1.92/0.17 | 2.31/0.18 | 1.63/0.09 |
| rope | 1.46/0.29 | 1.28/0.25 | 1.42/0.29 | 1.61/0.12 |
| tire | 1.53/0.25 | 1.35/0.20 | 1.47/0.22 | 1.44/0.11 |
| Mean | 1.80 /0.29 | 1.61/0.22 | 1.85/0.24 | 1.72/0.14 |

Table 4. Comparison of anomaly localization and detection performance on MVTec3D [2]. RGB: only RGB images, D: only depth images, +: mean of RGB and depth predictions.

| | Method | Pixel-level | | | Image-level | | |
|---|---|---|---|---|---|---|---|
| | | AUC | AP | $F_1$-max | AUC | AP | $F_1$-max |
| RGB | DFMGAN[9] | 74.4 | 14.7 | 20.7 | 63.7 | 82.8 | 84.9 |
| | AnoDiffusion[11] | 91.2 | 22.8 | 29.6 | 71.7 | 87.1 | 86.6 |
| | AnomalyHybrid | 96.9 | 16.0 | 23.2 | 83.7 | 94.8 | 91.4 |
| D | AnomalyHybrid | 94.2 | 12.8 | 19.1 | 82.7 | 94.5 | 92.0 |
| + | AnomalyHybrid | 98.4 | 19.5 | 26.4 | 90.1 | 97.1 | 94.0 |

category as the base sets, and the other 2/3 from each category are combined as the test set. With the base sets, both AHD and AHE decoders generate 500 anomaly images for each category. These generated images are used for evaluating generation quality and training models for downstream tasks. We also construct lists of 100 sampled anomaly images from the testing dataset. To calculate LPIPS, we partition the generated 1,000 images into 100 groups by finding the lowest LPIPS. We compute the mean pairwise LPIPS within each group. The average LPIPS of all groups will be the final score. Table 1 and Table 3 show the comparisons of RGB-level anomaly generation on MVTecAD [1] and MVTec3D [2]. Table 3 also demonstrates the depth-level anomaly generation performance of AnomalyHybrid. Comparing to both GAN-based and Diffusion-based SOTA, AnomalyHybrid generates RGB-level anomalies with both the highest realistic and diversity on all evaluated datasets. On MVTec3D [2], the generated depth-level anomalies achieve 1.72 IS score that is higher than the AnoDiffusion's RGB-level anomaly generation performance. As visualized in Figure 6, AnomalyHybrid not only generates different level of anomalies but also diverse normal samples.

**Anomaly detection.** Table 2 and Table 4 illustrate the benefit of our generated images for downstream anomaly classification, detection and segmentation tasks on MVTecAD [1] and MVTec3D [2]. On MVTecAD [1] dataset, we also evaluate the contribution of using generated normal samples for training anomaly detectors. Overall, the classifier ResNet34 trained on images generated from both AHD and AHE decoders of AnomalyHybrid achieves the highest accuracy 52.6 comparing to the GAN-based SOTA. With the same anomaly detector UNet, our generated images bring the highest performance both in image-level and pixel-level anomaly detection. By using normal images generated by AnomalyHybrid, the detector gains 0.4 percentage improvement in pixel-level AP. On MVTec3D [2] dataset, we conduct experiments of using RGB-level anomalies, depth-level anomalies and both of them to train UNet. Under these three settings, AnomalyHybrid all achieves better performance than GAN-based SOTA. By using both RGB-level and depth-level anomalies, AnomalyHybrid gains around 4 percentages improvement in pixel-level anomaly localization and 6 percentages improvement in image-level anomaly detection.

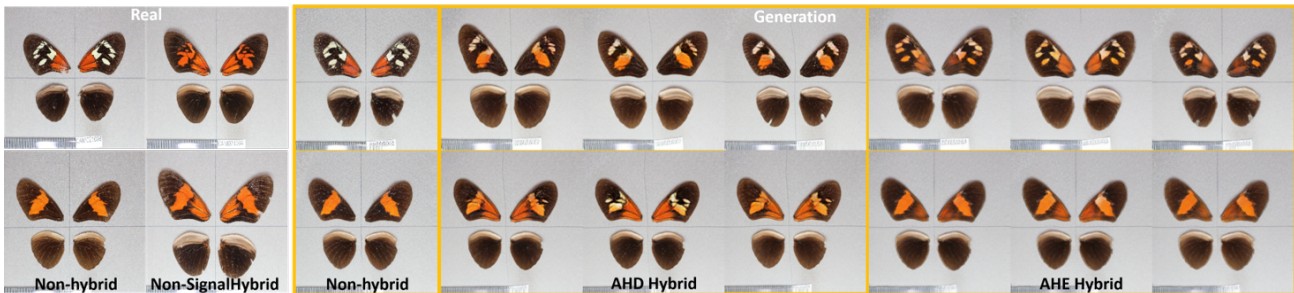

Figure 7. **Examples of anomaly generation by AnomalyHybrid on HeliconiusButterfly [4].**

## 4.4. Ablation

**Decoders.** As shown in Table 5, on MVTecAD [1], AHD decoder achieves higher IS/LPIPS scores that indicate more diverse anomalies than AHE decoder. However, the classifier(ResNet34) trained on anomalies generated by the AHD decoder achieves 1 percentage lower accuracy than the AHE decoder's. The reason is that AHD decoder focuses on less texture details than AHE decoder does. By using anomalies generated by both AHD and AHE decoders, the classifier gains 5.2 percentages accuracy improvement.

Table 6 illustrates the comparison of AHD and AHE anomalies for classification on HeliconiusButterfly [4] dataset. We use DINOv2 [19] to extract image features and simply use SGD and a 3 linear layers head as the detectors. The trainset contains only SignalHybrid and non-hybrid images. The testset consists of 3-type hybrids, including SignalHybrid, Non-signalHybrid and MimicHybrid. Figure 7 demonstrates 2 types of non-hybrid and 2 types of non-signal hybrid. Since the MimicHybrid is similar to the SignalHybrid, the classifier directly trained on the original trainset achieves the best performance on this type. The baseline anomaly generation method Anomaly-Factory [38], having only AHE branch, improves the classification performance more than 10 percentages. Different with AnomalyFactory [38], our network consists of both AHD and AHE branches that generate diverse and authentic hybrid as shown in Figure 7. With the help of anomalies generated by AHE decoder, the classifier achieves the highest harmonic mean recall 0.551 on 3-type hybrids.

**Manipulation.** We evaluate the effectiveness of different manipulations for anomaly generation and classification on Hazelnut of MVTecAD [1]. Table 7 shows the performance of using different manipulations. As shown in Figure 5, there are four types of defects and three out of them are close to depth values changing. The Remove manipulation always generates hole-like defect and causes ambiguity in other types, such as cut. Therefore, it achieves the lowest performance in diverse generation and defect classification. On the contrary, the Merge and Replace manipulations generate anomalies similar to the original types but with higher diversity. They both achieves the second highest classification accuracy. By randomly applying different manipu-

Table 5. Ablation study of decoders for anomaly generation and classification performance on MVTecAD [1].

| Decoder | | Generation | | Classification |
|---|---|---|---|---|
| AHD | AHE | IS↑ | LPIPS↑ | Accuracy↑ |
| - | v | 1.88 | 0.35 | 48.2 |
| v | - | 1.99 | **0.36** | 47.4 |
| v | v | **2.06** | 0.32 | **52.6** |

Table 6. Ablation study of anomaly generation and classification on HeliconiusButterfly [4]. (∗Without using manipulation.)

| Recall@ | Trainset | Baseline | **AHD** | **AHE** |
|---|---|---|---|---|
| | Classifier: Linear/SGD/Max(Linear, SGD) | | | |
| SignalHybrid | 0.847/**0.923**/0.893 | 0.764/0.792/0.789 | 0.781/0.778/0.784 | 0.811/0.860/0.855 |
| Non-SignalH | 0.143/0.143/0.036 | 0.214/0.357/0.357 | 0.250/0.357/0.357 | 0.321/**0.429**/0.429 |
| MimicHybrid | **0.621**/0.605/**0.621** | 0.435/0.431/0.419 | 0.524/0.484/0.500 | 0.480/0.509/0.516 |
| HMean | 0.306/0.308/0.098 | 0.363/0.470/0.465 | 0.417/0.488/0.494 | 0.467/0.549/**0.551** |

Table 7. Ablation study of manipulation for anomaly generation and classification performance on Hazelnut of MVTecAD [1].

| Manipulation | | | | Generation | | Classification |
|---|---|---|---|---|---|---|
| Merge | Remove | Replace | TPS | IS↑ | LPIPS↑ | Accuracy↑ |
| v | - | - | - | 1.716 | 0.320 | 81.6 |
| - | v | - | - | 1.668 | **0.327** | 53.1 |
| - | - | v | - | 1.723 | 0.321 | 81.6 |
| v | v | v | - | 1.746 | 0.321 | 75.5 |
| v | v | v | v | **2.163** | 0.326 | **88.5** |

lations, we gain 0.078 higher IS score for anomaly generation and 22.4 acc improvement for anomaly classification. With TPS manipluation, we increase the variety of anomaly shapes and achieve the overall highest performance.

## 5. Conclusion

In this paper, we propose a domain-agnostic framework, AnomalyHybrid, that generates diverse and authentic anomalies refer to multimodal conditional controls. It significantly optimizes existing GAN-based anomaly generation paradigm of learning multiple dedicated generative models per defect types. Extensive experiments conducted on three datasets demonstrate the superiority of Anomaly-Hybrid in general anomaly generation and the downstream anomaly detection tasks. With well-designed unsupervised training, AnomalyHybrid is easily generalized to other applications like edge extraction, depth estimation, and Out-of-Distribution detection. We believe that it will contribute more to downstream tasks with wilder extension.