# AnomalyHybrid: A Domain-agnostic Generative Framework for General Anomaly Detection

## Supplementary Material

## 6. Implementation

### 6.1. Datasets

**MVTecAD** [1] contains 10 object and 5 texture industrial products, such as bottle and leather. It consists of 3,629 normal images for training and 1,725 images for testing. There are 1,258 anomaly images of the testing set with pixel-level labelled various types of defects and the rest are normal images. Each class contains 60 to 320 color images with the resolution ranges from 700x700 to 1024x1024 pixels. In the testing set, defective appearance varies in different sizes, shapes and types, and most cases only contain a small fraction of anomalous pixels.

Figure 8 to Figure 10 visualize images generated by AnomalyHybrid and the predictions by a simple UNet trained on the generated images. AnomalyHybrid achieves good performance on 14 of 15 categories and fails only in the screw category.

**MVTec3D** [2] consists of 10 categories of industrial objects. It contains a total of 2,656 training samples, and 1,137 testing samples. Each sample has a colored point cloud that consists of a 3-channel tensor representing (x, y, z) coordinates and a 3-channel image with a resolution of 1920x1200 pixels. The 3D scans were acquired by an industrial sensor using structured light. The 41 types of typical anomalies either visualized in RGB images or Depth maps.

Figure 11 and Figure 12 visualize the real anomalies. Table 8 shows the detailed performance on each category. The images of the bagel and the cookie show cracks in the objects. The surfaces of the cableG and the dowel exhibit geometrical deformations. By combining the predictions of RGB and depth, AnomalyHybrid achieves good performance on 10 categories.

**HeliconiusButterfly** [4] is comprised of a subset of the Heliconius Collection (Cambridge Butterfly) [10] that is a compilation of images from Chris Jiggins' research group [1] at the University of Cambridge. It encompasses two aspects of biological development and evolutionary change, hybridization and mimicry. The training data comprises 2,084 images from all the Heliconius(H.) erato subspecies and the most common hybrid. The most common hybrid refers to a specific combination of the parent subspecies that has the most images. This hybrid is called the signal hybrid; other hybrids are called the non-signal hybrids. Heliconius(H.) melpomene mimics of the signal hybrid parent subspecies and their hybrids in H. erato. The two sub-

species of H. melpomene are those that mimic the parent subspecies of the signal hybrid of H. erato. The visually different appearances among subspecies in different regions and visually mimicking appearances between species in the same regions result in large intra-species variation within H. melpomene (or H. erato) and small inter-species variation between H. melpomene and H. erato.

Table 9 visualizes samples of 14 non-hybrid and 1 signal hybrid categories in trainset. As shown in Table 10 and Table 11, the test set totally comprises 2,350 images. Most of them are from 14 H. erato subspecies, 1 signal hybrid, and 5 non-signal hybrids. It also contains images from two subspecies of H. melpomene and their hybrid. Figure 13 illustrates the realistic images generated by AnomalyHybrid comparing to the real non-hybrid and hybrid butterfly images.

### 6.2. Training details

We extract edge and depth maps with Pidinet [22] and DepthAnythingV2 [26] on original images. We rescale all inputs, images, depth and edge maps, to a resolution of 256x256. The local TPS augmentation percentage is 0.99. The Adam optimizer has an initial learning rate of 2e-4 and decreases the learning rate with linear schedule. With sets of image, depth and edge, we firstly train RGB-level anomaly generator(AnomalyHybrid) 250 epochs from scratch with a batch size of 28 on four NVIDIA GeForce RTX 3080 Ti GPUs. Then, we use the pre-trained AnomalyHybrid model to initialize depth-level anomaly generator(AnomalyHybrid*) and further train it for 100 epochs with same setting.

### 6.3. Model size comparison

The GAN-based architecture of AnomalyHybrid has a 769MB-sized generator and a has 21MB-sized discriminator. The inference time on 256x256 image is around 0.287s. Most existing methods [9, 11, 33] focus on learning a dedicated generative model for each dataset and M particular anomaly predictors for M categories. AnomalyDiffusion [11] learns a 1.33GB-sized[2] diffusion model and RealNet [33] trains a 2.1GB-sized[3] diffusion model on MVTecAD[1] dataset for anomaly generation. DFMGAN [9] learns a 305.2MB-sized[4] GAN model only for the hole defect of Hazelnut of the MVTecAD 15 categories.

---

[1] https://doi.org/10.5281/zenodo.2549523

[2] https://github.com/sjtuplayer/anomalydiffusion
[3] https://github.com/cnulab/RealNet
[4] https://github.com/Ldhlwh/DFMGAN

Table 8. Comparison of anomaly localization and detection performance using RGB-level and Depth-level anomalies on MVTec3D [2].

| Category | RGB | Depth | RGBD | RGB | Depth | RGBD |
|---|---|---|---|---|---|---|
| | **Pixel-level** AUC/AP/$F_1$-max | | | **Image-level** AUC/AP/$F_1$-max | | |
| bagel | 97.7/5.4/11.5 | 94.1/4.4/10.9 | **98.5**/**6.2**/**14.1** | 91.5/97.4/95.6 | 89.9/96.9/94.7 | **96.9**/**99.1**/**97.4** |
| cableG | **95.1**/**4.8**/11.4 | 87.5/0.8/4.4 | 94.5/4.5/**12.4** | **99.0**/**99.8**/**97.3** | 65.2/89.1/90.2 | 92.9/98.2/93.2 |
| carrot | 99.4/21.5/26.9 | 98.2/23.5/33.7 | **99.7**/**29.5**/**36.3** | 90.0/97.9/93.3 | 98.3/99.7/**98.3** | **98.9**/**99.8**/97.7 |
| cookie | 90.6/**33.9**/35.2 | 96.8/27.5/34.1 | **98.5**/33.0/**37.6** | 88.0/96.9/89.7 | 95.9/98.9/94.6 | **97.6**/**99.3**/**96.2** |
| dowel | 98.7/**39.6**/**44.7** | 92.4/12.7/20.6 | **99.1**/27.1/32.2 | **89.3**/**97.2**/**92.1** | 66.5/90.1/88.7 | 82.2/95.4/89.3 |
| foam | 95.2/**7.2**/**17.7** | 87.7/0.5/3.3 | **96.7**/5.2/14.7 | 87.9/96.9/**92.9** | 82.4/95.8/88.9 | **89.9**/**97.4**/92.7 |
| peach | 98.7/21.6/30.4 | 97.1/19.1/25.2 | **99.2**/**26.1**/**33.9** | 87.5/96.5/93.0 | 88.6/97.2/91.5 | **95.8**/**99.0**/**94.4** |
| potato | 97.3/5.2/12.2 | **99.4**/30.9/**36.1** | 99.4/**31.1**/33.9 | 48.2/79.1/89.6 | **80.2**/**93.2**/**92.8** | 76.3/92.8/92.3 |
| rope | 99.0/6.2/**14.5** | 95.5/5.7/12.9 | **99.3**/**6.7**/14.4 | 94.4/97.9/92.9 | 95.2/97.9/92.0 | **99.0**/**99.6**/**97.7** |
| tire | **99.2**/24.9/29.2 | 92.7/2.9/9.6 | 98.9/**25.4**/**34.5** | 81.7/**93.9**/**91.4** | 65.1/86.4/88.7 | 71.6/90.4/88.7 |
| Mean | 97.1/17.0/23.4 | 94.2/12.8/19.1 | **98.4**/**19.5**/**26.4** | 85.7/95.3/92.8 | 82.7/94.5/92.0 | **90.1**/**97.1**/**94.0** |

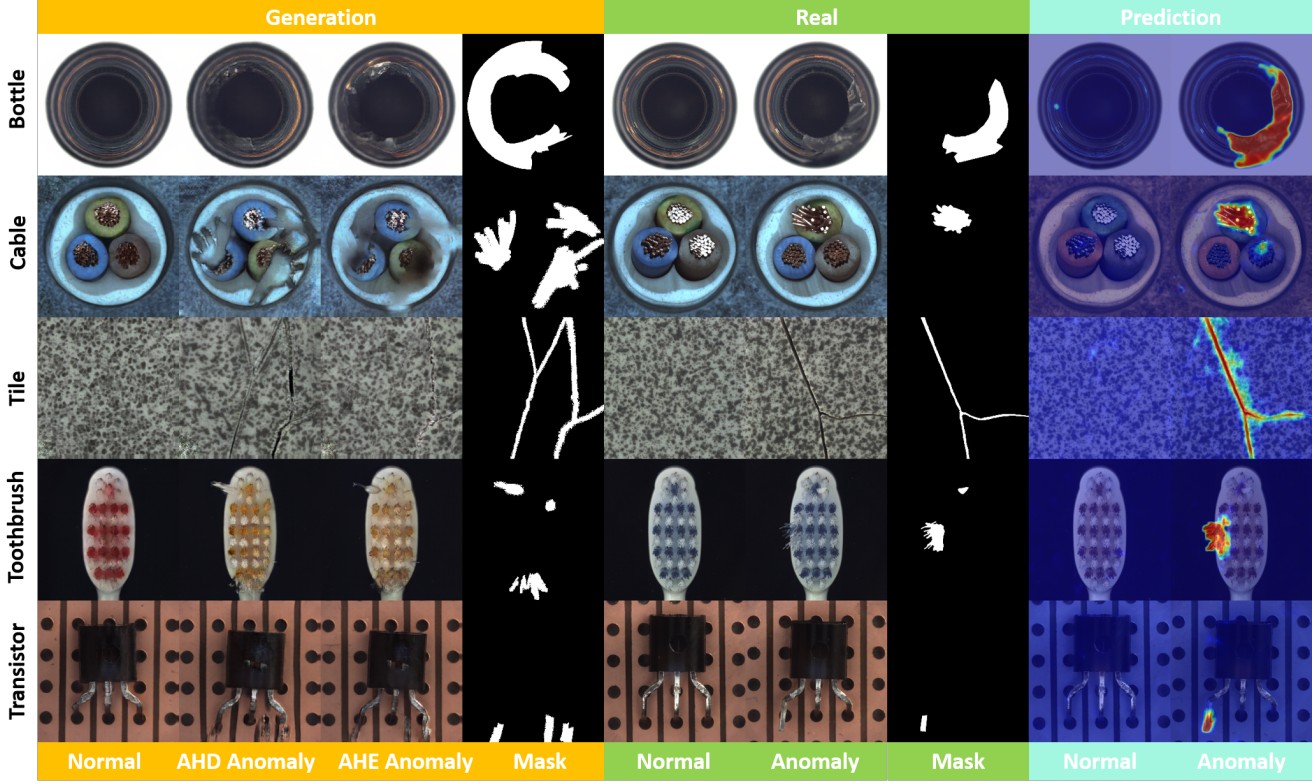

Figure 8. **Visualization of anomaly generation and detection on 5/15 categories of MVTecAD [1].**

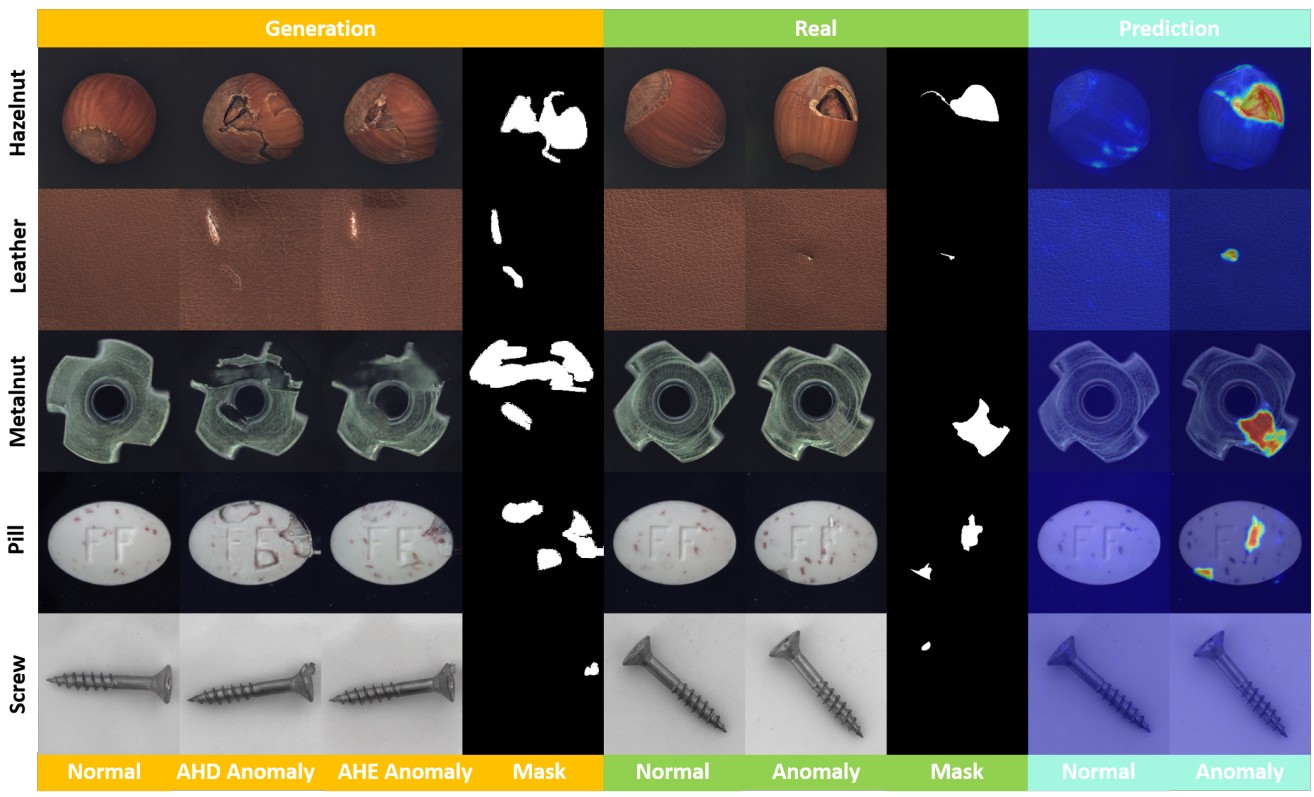

Figure 9. **Visualization of anomaly generation and detection on 5/15 categories of MVTecAD [1].**

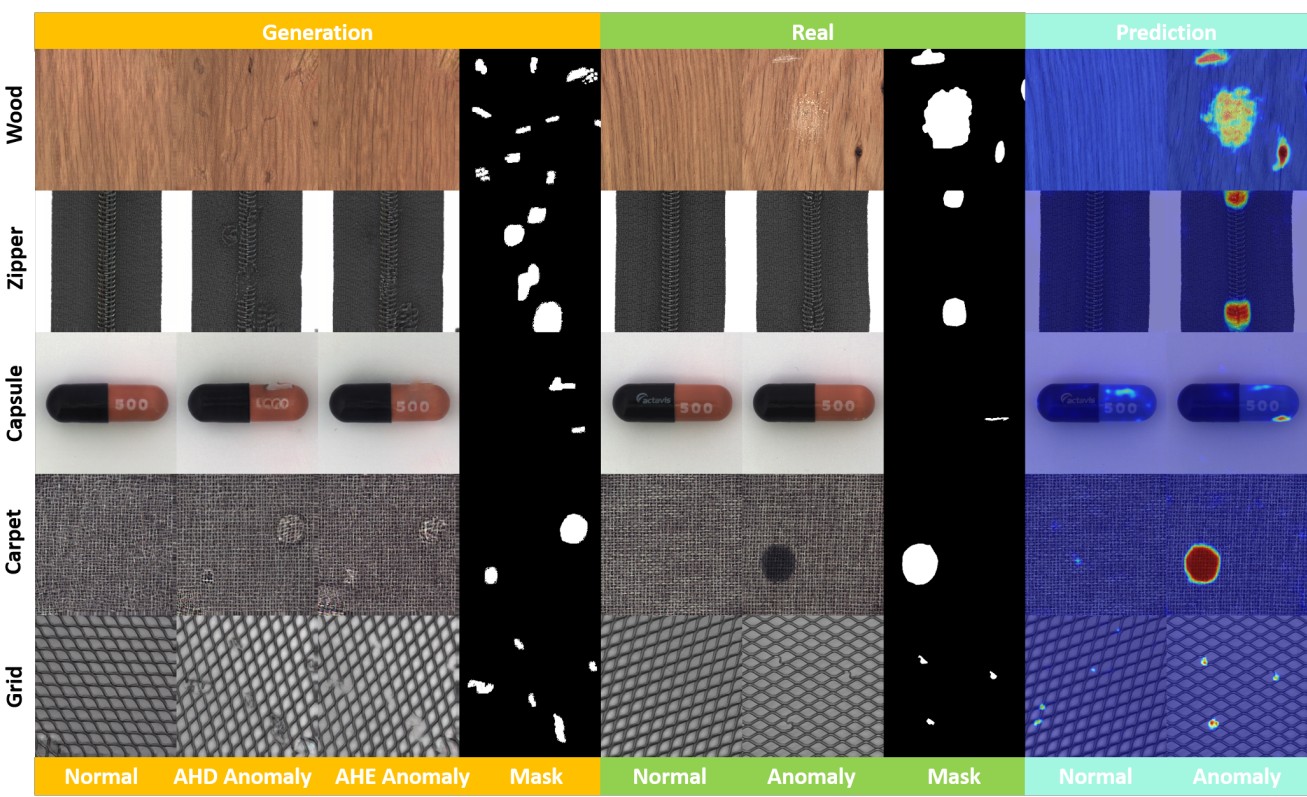

Figure 10. **Visualization of anomaly generation and detection on 5/15 categories of MVTecAD [1].**

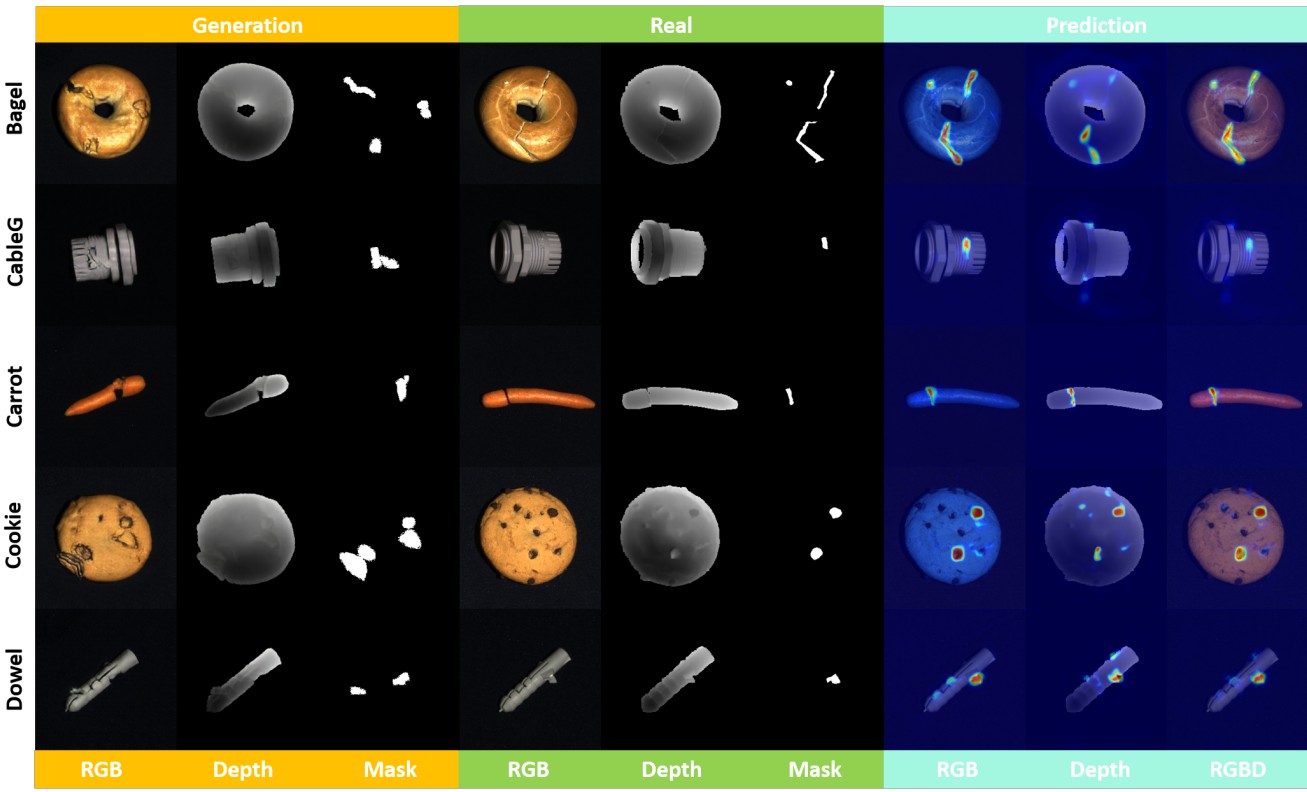

Figure 11. **Visualization of anomaly generation and detection on 5/10 categories of MVTec3D [2].**

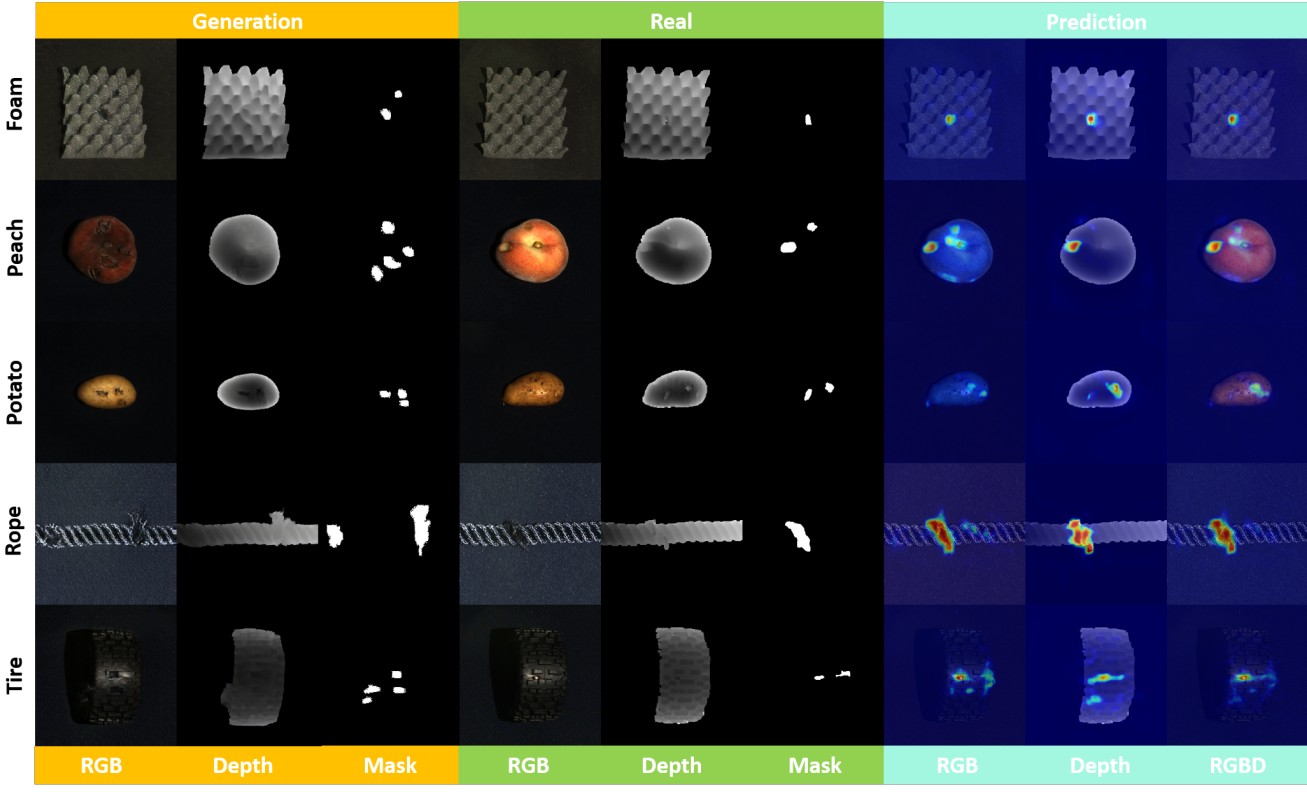

Figure 12. **Visualization of anomaly generation and detection on 5/10 categories of MVTec3D [2].**

Table 9. Statistic of **Non-Hybrid** and **Signal-Hybrid** in **trainset** of HeliconiusButterfly [4].

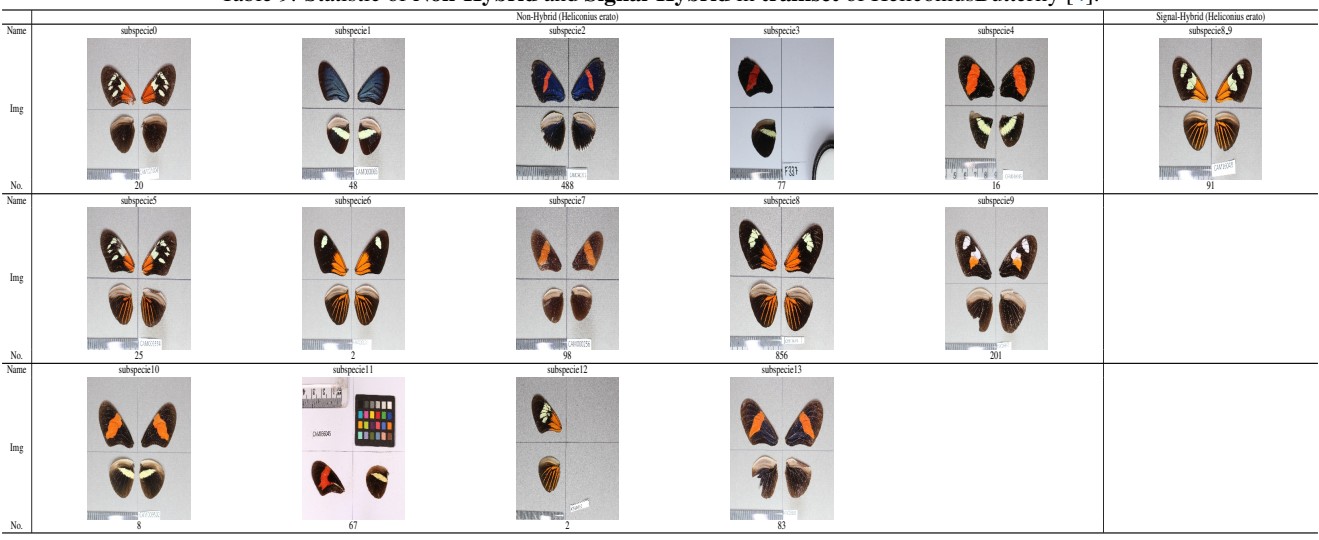

Table 10. Statistic of butterfly **Non-Hybrid** in **testset** of HeliconiusButterfly [4].

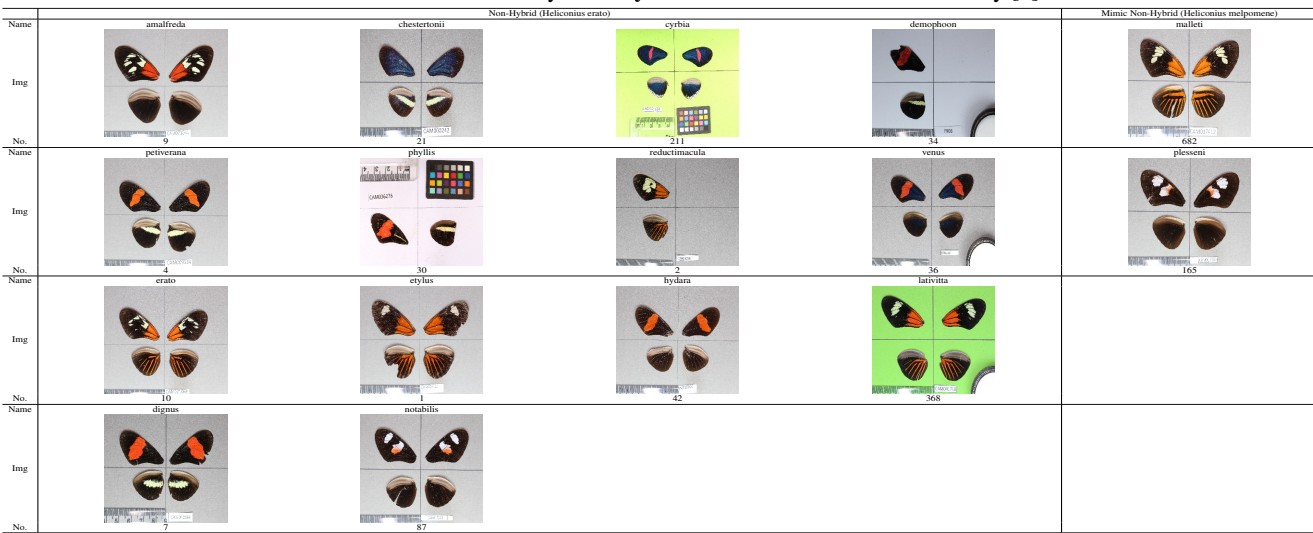

Table 11. Statistic of butterfly **Hybrid** in **testset** of HeliconiusButterfly [4].

| | Signal-Hybrid (Heliconius erato) | Non-Signal-Hybrid (Heliconius erato) | | | | | Mimic Hybrid (Heliconius melpomene) |
|---|---|---|---|---|---|---|---|
| Name | notabilis_lativitta | chestertonii_venus | venus_chestertoni | hydara_amalfreda | hydara_erato | hydara_petiverana | plesseni_malleti |
| Img | | | | | | | |
| No. | 365 | 1 | 1 | 8 | 1 | 17 | 248 |

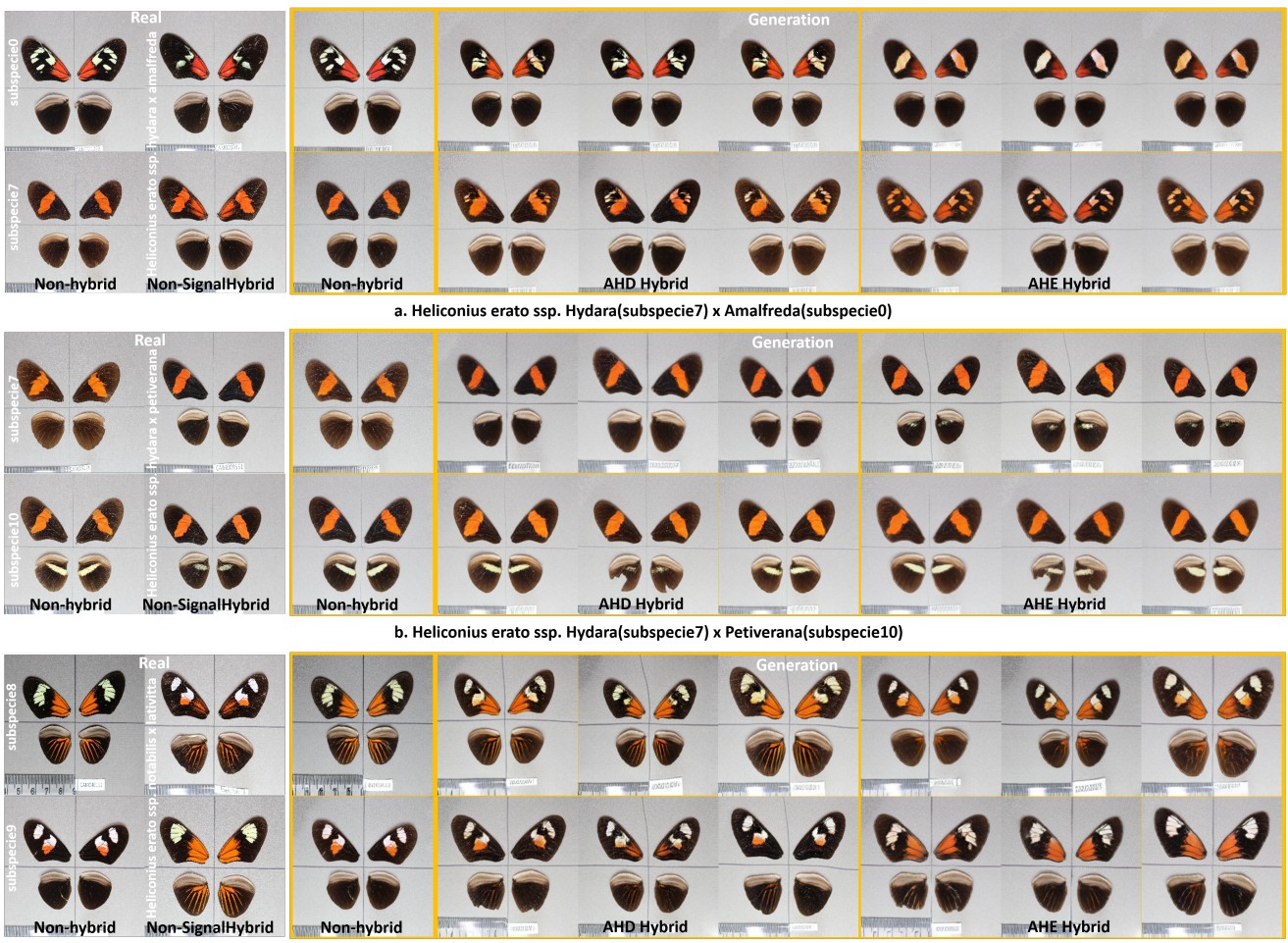

a. Heliconius erato ssp. Hydara(subspecie7) x Amalfreda(subspecie0)

b. Heliconius erato ssp. Hydara(subspecie7) x Petiverana(subspecie10)

c. Heliconius erato ssp. Notabilis(subspecie9) x Lativitta(subspecie8)

Figure 13. **Visualization of butterfly hybrid and non-hybrid generation on HeliconiusButterfly [4]. (\*The corresponding informa-tion of subspecie names in trainset and testset is used only for illustration but not in generation.)**