# OpenReview forum: "AnomalyHybrid: A Domain-agnostic Generative Framework for General Anomaly Detection"
_thecvf.com/CVPR/2025/Workshop/SyntaGen — SyntaGen 2025 Poster_

### Official Review · Reviewer_yUQs · 2025-03-27

**Rating:** 7
**Confidence:** 4

**Review:**

**Paper Summary**:

This paper proposes a generation framework which can synthesize diverse and authentic anomaly samples across domain application with controllable signals such as color, depth, and edge information. To achieve this, a GAN-based model with two decoders is employed to incorporate appearance of reference image into depth and edge structure of target image respectively. Extensive experiments on several anomaly benchmarks are provided to demonstrate that the proposed method could surpass recent gan-based state-of-the-art results.

**Strengths**:
- As shown in Fig.3 , the proposed anomaly generation framework seems to be more powerful than previous frameworks where it could cover all four essential features in anomaly generation problem.
- Extensive experimental results shown in Tab. 1 and Tab. 2 demonstrate the effectiveness of the proposed method where it could mostly surpass existing methods with a large margin across different benchmarks and metrics.

**Weaknesses**:
- The rationale and how to extract the generated fusion map $M_{gen}$ are not clearly presented.
- Additional discussion about the rationale of choosing the base generative model for the anomaly generation framework (GAN-based in this case) would be helpful to further understand the advantage of using GAN instead of diffusion-based model.

In general, I think this work addresses an interesting problem in anomaly generation. The proposed method is beneficial compared to previous framework since it could align different levels of anomalies across versatile application domains. In addition, the experimental results demonstrate a significant improvement compared to previous methods. Hence, I recommend an acceptance for this work.

---

### Official Review · Reviewer_Zndp · 2025-03-27

**Rating:** 6
**Confidence:** 4

**Review:**

**Summary**

This paper proposes using conditional GANs to synthesize data for anomaly detection, classification, and localization tasks. The GAN models are conditioned on depth and edge information, enabling them to integrate depth and edge features with RGB images. This approach allows for the generation of diverse anomalies by manipulating depth and edge map. Additionally, the method enables anomaly generation in the depth domain, making it suitable for training depth-level anomaly detection models. Experimental results demonstrate that using the synthesized images as a training set leads to superior performance compared to other data generation methods on both industrial and biological test datasets.

**Strengths:**
- This paper leverages GANs to address a compelling and challenging problem.
- The use of different conditioning modes influences various aspects of image generation, leading to greater diversity—not just within an image level (e.g., through random noise).
- Extensive experiments conducted on diverse test sets ensure comprehensive evaluation, covering different aspects of the problem.

**Weaknesses:**
- Some parts of the explanation are unclear. For example, it is not explicitly stated whether real image $I_{r}$, $I_{t}$ refer to the input image or an augmented version. Additionally, the reasoning behind applying augmentation to both the RGB and conditional branches is not well explained.
- The definition of "authentic anomaly" is somewhat ambiguous. Since the generated structure is based on the conditioned edge or depth, which is itself an augmented version of the real condition, it is unclear how this augmentation guarantees authenticity. Similar concerns arise regarding the manipulation technique.
- The evaluation tables contain dense numerical data, making them difficult to read. Reorganizing or adjusting the presentation for better clarity would be beneficial.

This work addresses the interesting challenge of anomaly detection, including depth-level anomaly detection, using a simple conditional GAN. Despite the use of GANs and relatively straightforward anomaly detection models, the proposed approach significantly enhances performance, highlighting the effectiveness of the techniques introduced in this study. Given its contributions and demonstrated improvements, I recommend accepting this work.

---

### Official Review · Reviewer_76Wn · 2025-03-27
**Review for paper 9**

**Rating:** 5
**Confidence:** 2

**Review:**

**Summary**

The paper tackles the problem of generating data for anomaly detection. The authors proposed a unified pipeline that can generate anomaly samples for many different industries. By utilizing depth and edge from off-the-shelf models, the authors trained a GAN framework (AnomalyHybrid) to generate anomalies. The authors also proposed an unsupervised training framework by using augmentations on the reference image/depth/edge to create pseudo anomaly image/depth/edge triplets.

**Strength**

- The idea of adding depth and edge is straightforward and convincing. This enhances anomaly generation especially in low-data scenarios by providing richer structural cues.
- The results in Table 1 show that AnomalyHybrid generates samples that are closer to the distribution of data compared to other GAN-based methods. This could be due to the ability to leverage depth and edge as inputs.

**Weaknesses**

- The current versions of Tables 1, 2, and 3 are overly dense and difficult to read. The authors should consider redesigning these tables with better spacing and clearer annotation. In particular, a detailed caption explaining the meaning of highlighted numbers (e.g., bold for best results, gray for second best) would help readers interpret the performance metrics.
- Unclear on how unsupervised training is utilized. The writing of the paper is a bit unclear here, is the Augmentation described in line 309 used to convert a normal image/depth/edge into a pseudo anomaly image/depth/edge triplet? If this is true, how can this work on a dataset like HeliconiusButterfly, since on this dataset, the anomalies are not visual distortions.
- The paper writing is unclear and could be improve. For the losses section (line 323) it is unclear what the reference and target images are.

---

### Decision · Program_Chairs · 2025-03-30

**Decision:**

Accept (Poster)

**Comment:**

This paper received mixed reviews, ranging from borderline rejection to weak acceptance. It introduces a compelling anomaly generation framework that effectively utilizes depth and edge information to enhance sample quality, particularly in low-data scenarios. Extensive experiments across diverse benchmarks demonstrate notable performance improvements over existing GAN-based methods, underscoring the robustness and effectiveness of the proposed approach. However, the paper is hindered by unclear writing—especially in explaining unsupervised training, loss definitions, and augmentation strategies. The evaluation tables are overly dense, and important design choices, such as the preference for GANs over diffusion models, are insufficiently justified. After careful consideration, the AC recommends acceptance, with the expectation that the authors will address the reviewers’ comments and improve the clarity and presentation in the final version.